elLIFE

# A long non-coding RNA targets microRNA miR-34a to regulate colon cancer stem cell asymmetric division

Lihua Wang[1†], Pengcheng Bu[2,3,4†], Yiwei Ai[2,3], Tara Srinivasan[3], Huanhuan Joyce Chen[5], Kun Xiang[3], Steven M Lipkin[6*], Xiling Shen[1,2,3,4*]

[1]Department of Biological and Environmental Engineering, Cornell University, Ithaca, United States; [2]School of Electrical and Computer Engineering, Cornell University, Ithaca, United States; [3]Department of Biomedical Engineering, Cornell University, Ithaca, United States; [4]Department of Biomedical Engineering, Duke University, Durham, United States; [5]Meyer Cancer Center, Weill Corenll Medical College, New York, United States; [6]Deparments of Medicine, Genetic Medicine and Surgery, Weill Cornell Medical College, New York, United States

*For correspondence: stl2012@
med.cornell.edu (SML); xs37@
duke.edu (XS)

[†]These authors contributed equally to this work

Competing interests: The authors declare that no competing interests exist.

**Abstract** The roles of long non-coding RNAs (lncRNAs) in regulating cancer and stem cells are being increasingly appreciated. Its diverse mechanisms provide the regulatory network with a bigger repertoire to increase complexity. Here we report a novel LncRNA, Lnc34a, that is enriched in colon cancer stem cells (CCSCs) and initiates asymmetric division by directly targeting the microRNA miR-34a to cause its spatial imbalance. Lnc34a recruits Dnmt3a via PHB2 and HDAC1 to methylate and deacetylate the miR-34a promoter simultaneously, hence epigenetically silencing miR-34a expression independent of its upstream regulator, p53. Lnc34a levels affect CCSC self-renewal and colorectal cancer (CRC) growth in xenograft models. Lnc34a is upregulated in late-stage CRCs, contributing to epigenetic miR-34a silencing and CRC proliferation. The fact that lncRNA targets microRNA highlights the regulatory complexity of non-coding RNAs (ncRNAs), which occupy the bulk of the genome.

## Introduction

A downstream target of p53, the microRNA miR-34a is a well-known tumor suppressor in various types of cancers (*Chang et al., 2007*; *He et al., 2007*). Among its many functions, miR-34a has been shown to limit self-renewal of cancer stem cells (*Bu et al., 2013*; *Liu et al., 2011*). miR-34a mimics such as MRX34 are among the first microRNA mimics to reach clinical trial for cancer therapy (*Bader, 2012*; *Bouchie, 2013*). Besides cancer, miR-34a has been shown to regulate stem cell differentiation, somatic stem cell reprogramming, cardiac ageing, neurodegeneration, ciliogenesis, bone resorption, and metabolism (*Aranha et al., 2011*; *Boon et al., 2013*; *Choi et al., 2011*; *Krzeszinski et al., 2014*; *Liu et al., 2012*; *Song et al., 2014*; *Xu et al., 2015*).

Loss of p53 function can lead to downregulation of miR-34a; however, miR-34a expression also tends to be silenced due to aberrant CpG methylation of its promoter in many types of cancer, including breast, prostate, lung, colon, kidney, bladder, pancreatic, and ovarian cancer (*Corney et al., 2010*; *Kong et al., 2012*; *Lodygin et al., 2008*). Methylation of the miR-34a promoter is positively correlated with and miR-34a expression is inversely correlated with progression of colorectal cancer (CRC) (*Siemens et al., 2013*). However it is completely unclear how miR-34a was silenced by epigenetic modification.

**eLife digest** Tumors are made of millions of cells that are not all the same. A type of cancer cell known as cancer stem cells (or CSCs for short) are often better at dividing to produce new cells and moving to new sites in the body than other types of cancer cell. Very small molecules called micro ribonucleic acids (or microRNAs for short) can influence how CSCs grow and divide by regulating the activity of specific genes. For example, a microRNA molecule called miR-34a suppresses the activity of several genes – which slows the growth of various tumors, including lung and bowel cancers. This miR-34a is often missing from some types of cells in advanced tumors.

Genes encode the instructions to produce RNAs, and Wang, Bu et al. wanted to find out what stops miR-34a being produced in certain bowel cancer cells. The experiments revealed a new, very long RNA molecule – named long non-coding RNA 34 (or Lnc34a) – that binds to the gene that encodes miR-34a. Lnc34a recruits proteins that modify the gene and switch off the production of miR-34a.

Furthermore, microscopy experiments revealed that when colon cancer cells divide, Lnc34a is distributed unevenly so that it blocks the production of miR-34a in one daughter cell but not the other. Lastly, Wang, Bu et al. confirmed that Lnc34a is found in higher levels in CSCs than in other cancer cells, which helps them to grow and divide more rapidly. Future experiments will try to find out what controls the production of Lnc34a and search for drugs that can block this process in cancer cells.

Normal stem cells often divide asymmetrically to produce one daughter cell like itself for self-renewal and another daughter cell unlike itself to go down a path of differentiation (*Neumuller and Knoblich, 2009*). Asymmetric division allows stem cells to maintain self-renewal while generating a heterogeneous population for cellular diversity (*Reya et al., 2001*). Tumor cells are usually heterogeneous and have a wide range of potential for tumorigenesis, proliferation, and metastasis. Recent studies have reported that cancer cells, including colorectal, glioma, lung and breast cancer cells, could also divide asymmetrically, generating progenies with different proliferation capabilities (*Bu et al., 2013*; *Dey-Guha et al., 2011*; *Lathia et al., 2011*; *O'Brien et al., 2012*; *Pece et al., 2010*; *Pine et al., 2010*; *Sugiarto et al., 2011*). The frequencies of symmetric vs. asymmetric divisions are associated with cancer proliferation and progression. Disruption to asymmetric division in favor of symmetric self-renewal alters the balance between self-renewal and differentiation, which has been linked to neoplastic transformation and tumor growth (*Cicalese et al., 2009*; *Sugiarto et al., 2011*).

Here, we discovered that a novel lncRNA, Lnc34a, directly targets the miR-34a promoter for epigenetic silencing by recruiting the DNA methyltransferase Dnmt3a via Prohibitin-2 (PHB2) and Histone Deacetylase 1 (HDAC1). Asymmetric distribution of Lnc34a during colon cancer stem cell (CCSC) division leads to asymmetric daughter cell fate. Its suppression leads to differentiation while its abundance leads to CCSC proliferation via symmetric self-renewal. Lnc34a tends to be upregulated in late-stage CRC, associated with miR-34a silencing. The ability of lncRNA to target microRNA provides RNA circuitry more ways to increases the complexity of the regulatory network.

## Results

### A lncRNA overlapping with miR-34a promoter

We performed RT-PCR with 10 pairs of primers to scan for potential transcripts overlapping the miR-34a promoter and its downstream sequence. A 293 base pair (bp) transcript fragment was amplified. Rapid amplification of cDNA ends (RACE) further identified a full-length, 693 bp transcript (*Figure 1—figure supplement 1A,B*). Northern blot confirmed the existence and size of the transcript in seven CRC cell lines and two colon cancer stem cell (CCSC) lines (*Figure 1A,B*, *Figure 1—figure supplement 1D*). The CCSCs were isolated from two early-stage CRC specimen, and were functionally validated by serial sphere formation, tumor initiation, and marker staining (*Bu et al., 2013*). The original frozen stocks from the first passage were used in the study. The transcript is composed of

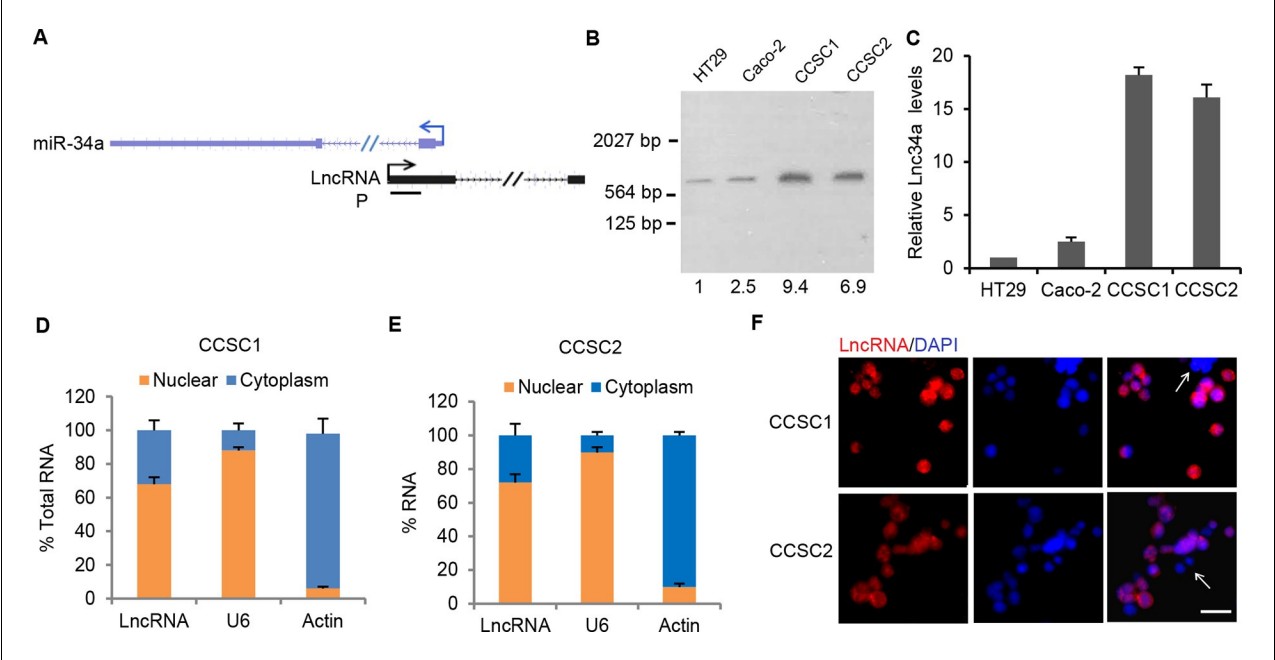

**Figure 1.** Characterization of Lnc34a. (**A**) Schematic illustration of Lnc34a (shown in black) and miR-34a (shown in blue) gene structure. Lnc34a and miR-34a contain two exons and are transcribed in different directions. P, probe for Northern blot in (**B**). (**B**) Northern blot detection of Lnc34a with the probe shown in (**A**), quantified by Image J. (**C**) RT-qPCR detection of Lnc34a expression in colon cancer stem cells (CCSC1 and CCSC2) and well-established colon cancer cell lines (HT29 and Caco-2). (**D, E**) RT-qPCR detection of Lnc34A level in cellular fractions from CCSC1 (**D**) and CCSC2 (**E**) sphere cells. U6 and actin are the nuclear and cytoplasm controls, respectively. (**F**) Lnc34a expression in CCSC sphere cells detected by RNA-FISH. Scale bar, 20 μm.

The following source data and figure supplements are available for figure 1:

**Source data 1.** Information of CRC patients.

**Figure supplement 1.** Identification of Lnc34a.

**Figure supplement 2.** RNA FISH specificity and Lnc34a knockdown efficiency.

two exons, spanning nearly 15.3 kilobases (kb), and does not contain a valid Kozak sequence. The full-length transcript has no protein coding potential according to the Coding Potential Calculator (CPC) and Coding Potential Assessment Tool (CPAT) (*Kong et al., 2007*; *Wang et al., 2013*). We named the transcript Lnc34a.

To analyze Lnc34a expression in CRC cells, RT-qPCR was performed in 9 commonly used CRC cell and the two CCSC lines. Consistent with the Northern blot measurement, Lnc34a levels were significantly higher in the CCSC sphere cells (*Figure 1C*, *Figure 1—figure supplement 1C*). Cellular fractionation assays show enrichment of Lnc34a in the nuclear fraction (*Figure 1D,E*), and RNA fluorescence in situ hybridization (RNA FISH) indicates that Lnc34a is mainly in the nucleus (*Figure 1F*). RNA FISH specificity was validated when the same RNA-FISH probe did not detect Lnc34a after Lnc34a was knocked down by lentiviral short-hairpin RNA (shRNA) vectors in CCSC spheres (*Figure 1—figure supplement 2A*).

## Lnc34a asymmetry

Notably, RNA-FISH showed that a small population among the CCSC sphere cells did not express Lnc34a, although the majority did (*Figure 1F*). We then separated the sphere cells into two populations based on the expression levels of ALDH1, a CCSC marker (*Huang et al., 2009*). Flow analysis confirmed that ALDH1+ cells also express high levels of CD133, another CCSC marker (*Figure 2—figure supplement 1A*). RT-qPCR showed that, in both sphere cultures (CCSC1 and CCSC2), ALDH1 + cells have much higher Lnc34a expression levels than the ALDH1- cells (*Figure 2A,B*). We then

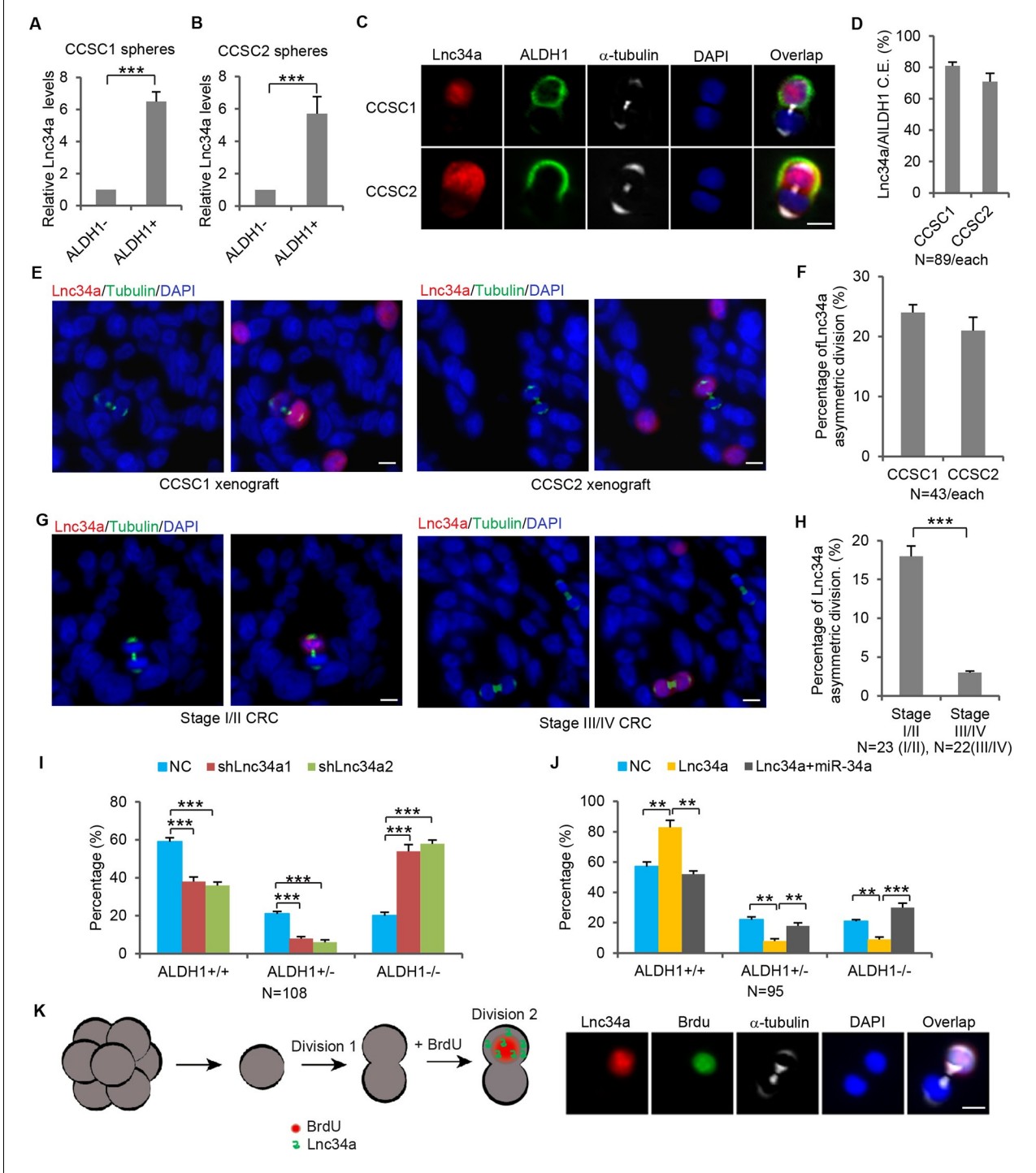

**Figure 2.** Lnc34a Asymmetry in CCSC division. (**A, B**) RT-qPCR detection of lnc34a in ALDH1+ and ALDH1- populations isolated from spheres of two independent patient-derived lines, CCSC1 (**A**) and CCSC2 (**B**). Lnc34a is high in ALDH1+ (CCSC) but low in ALDH1- (non-CCSC) cells. (**C**) Representative images of Lnc34a distribution in dividing pairs. α-tubulin staining is consistent with the telophase (final phase of mitosis) configuration of microtubules – the midbody at the division plane during cytokinesis and asters at the poles. ALDH1 identifies the CCSC daughter. (**D**) Quantification of Lnc34a/ALDH1 co-expression (C.E.) in daughter compartments of dividing pairs as shown in (**C**). (**E**) Representative images of Lnc34a asymmetry in dividing pairs in xenograft tumors derived from CCSC1 and CCSC2. Dividing pairs are identified by tubulin staining. (**F**) Percentage of Lnc34a asymmetry in dividing pairs in CCSC xenografts as shown in (**E**). (**G**) Representative images of asymmetric and symmetric Lnc34a distribution in dividing pairs in early- and late-stage human CRC specimens. (**H**) Percentage of Lnc34a asymmetry in dividing pairs in human CRC specimens. (**I**) Effect of Lnc34a knockdown on mode of division based on ALDH1 staining of dividing cell pairs. Lnc34a knockdown decreased asymmetric (ALDH1+/ALDH1-)

*Figure 2 continued on next page*

*Figure 2 continued*

division and symmetric self-renewal (ALDH1+/ALDH1+), and increased differentiation (ALDH1-/ALDH-). (J) Effect of ectopic Lnc34a expression on mode of division. Ectopic Lnc34a increased symmetric self-renewal (ALDH1+/ALDH+), and reduced asymmetric division (ALDH1+/ALDH1-) and differentiation (ALDH1-/ALDH1-). The effect of ectopic Lnc34a expression was abrogated by ectopic miR-34a expression. (K) Pair-cell BrdU incorporation assay showing asymmetric proliferative potential. Left, schematic representation of the experimental approach. Single sphere cells were allowed to divide once (1st division). Cells were then treated with BrdU for 3 hr to label cells that were re-entering the 2nd division. Right, representative images showing that the Lnc34a+ cells were more proliferative and incorporated BrdU. Scale bar, 8 μm. Error bars denote s.d. of triplicates. *p<0.05; **p<0.01; ***p<0.001. p-value was calculated based on Student's t-test.

The following figure supplement is available for figure 2:

**Figure supplement 1.** CCSCs co-express ALDH1 and CD133.

performed the pair-cell assay by plating single cells and allowing them to progress through one cell division (*Bultje et al., 2009*). α-tubulin staining was used to identify dividing cells (*Figure 2C*). Co-staining revealed that Lnc34a was asymmetrically distributed and enriched in the ALDH1+ (CCSC) daughter cells, which were also CD133+ (*Figure 2C,D*, *Figure 2—figure supplement 1B,C*).

Lnc34a asymmetry in dividing cell pairs was confirmed in vivo by RNA-FISH and tubulin staining of xenograft tumors derived from subcutaneously injected CCSCs (*Figure 2E,F*). We investigated Lnc34a asymmetry in 23 early-stage (stage I/II) and 22 late-stage (stage III/IV) human CRC specimens (*Figure 1—source data 1*). Lnc34a asymmetry in dividing cell pairs is more strongly associated with early-stage CRC, while late-stage CRC mostly has symmetric Lnc34a levels in dividing pairs (*Figure 2G,H*).

To investigate whether Lnc34a regulates CCSC division symmetry, we first knocked down Lnc34a using lentiviral shRNAs, which have been reported to knock down certain nuclear lncRNAs efficiently (*Castel and Martienssen, 2013*; *Di Ruscio et al., 2013*; *Wang et al., 2015*; *Xing et al., 2014*). Among the five tested shRNAs against Lnc34a, two showed efficient suppression of Lnc34a (shLnc34a1 and shLnc34a2; *Figure 1—figure supplement 2B*). Lnc34a knockdown decreased asymmetric division while increasing symmetric, ALDH1-/ALDH1- division (*Figure 2I*). We then ectopically expressed Lnc34a using lentiviral vectors. Higher level of ectopic Lnc34a was detected in the nucleus than in the cytoplasm (*Figure 1—figure supplement 2C*). Ectopic Lnc34a expression also decreased asymmetric division, but increased symmetric, ALDH1+/ALDH1+ division instead (*Figure 2J*). The phenotype was rescued by ectopic miR-34a expression, suggesting that Lnc34a regulates symmetry through miR-34a (*Figure 2J*). The same trend was observed with CD133 staining (*Figure 2—figure supplement 1C,D*). Therefore, ectopic Lnc34a seems to promote symmetric CCSC self-renewal, while Lnc34a silencing promotes differentiation.

Pair-cell BrdU incorporation assay showed that, when cultured in proliferative medium (DMEM with 10% FBS), the Lnc34a+ daughter cell starts incorporating BrdU and enters into the next division immediately, whereas the Lnc34a- daughter cells does not incorporate BrdU (*Figure 2K*). Therefore, the Lnc34a+ daughter cell has higher proliferative capacity.

## Lnc34a enhances CCSC self-renewal and tumorigenesis

Serial sphere propagation assays were performed to evaluate the effect of Lnc34a on CCSC self-renewal. CCSCs containing a control vector exhibited stable sphere formation capability through 3 generations of sphere propagation. Lnc34a knockdown strongly suppressed sphere formation capability, which was completely lost after 3 generations of passage (*Figure 3A,B*). In contrast, ectopic Lnc34a expression increased sphere numbers and sizes significantly. Ectopic miR-34a abrogated the effect of Lnc34a on sphere formation regulation, suggesting that Lnc34a promotes CCSC self-renewal by targeting miR-34a (*Figure 3A,C*).

Next, we used the mouse xenograft model to examine whether Lnc34a influences tumor growth. All five mice in the control group (injected with sphere cells containing the control vector) developed tumors. However, only three mice injected with sphere cells expressing shLnc34a1 and two mice injected with sphere cells expressing shLnc34a2 formed tumors, which are smaller than those of the control group (*Figure 3D,E*). All 5 mice injected with sphere cells ectopically expressing Lnc34a developed tumors, which are notably bigger than those in the control group. Ectopic miR-34a

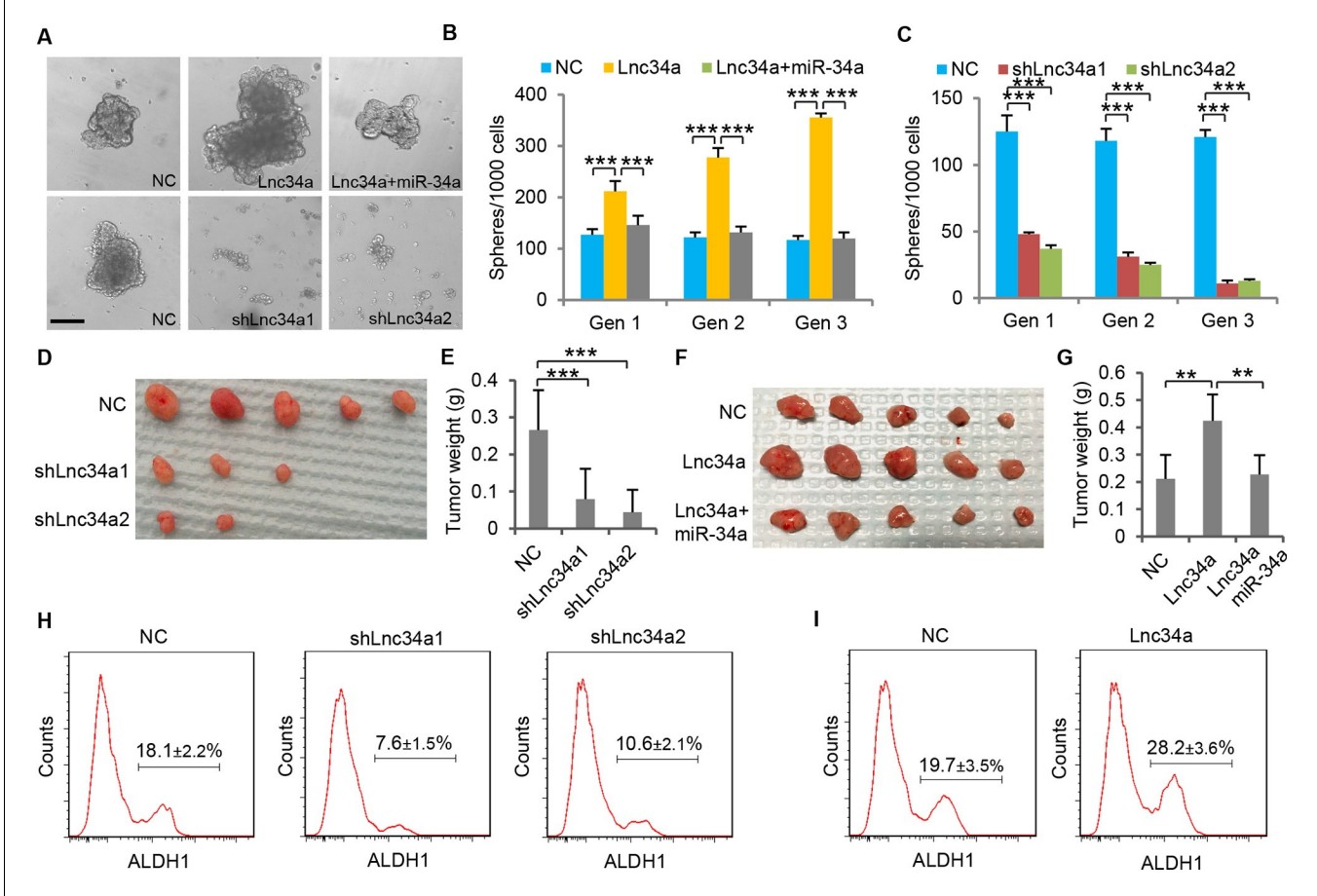

**Figure 3.** Lnc34a promotes CCSC self-renewal and tumor formation. (**A**) Representative images of CCSC spheres with Lnc34a knockdown (shLnc34a1 and shLnc34a2), ectopic Lnc34a expression (Lnc34a), and ectopic Lnc34a/miR-34a expression. (**B, C**) Sphere formation during serial passages after Lnc34a knockdown (**B**) and ectopic Lnc34a and miR-34a expression (**C**). Equal number of cells was passaged for 3 generations to form spheres. (**D, E**) Knockdown of Lnc34a (shLnc34a1 and shLnc34a2) reduced tumorigenicity, shown by images (**D**) and weights of xenograft tumors (**E**). (**F, G**) Ectopic Lnc34a expression (Lnc34a) enhances tumorigenicity, which can be abrogated by ectopic miR-34a expression. (**H, I**) FACS plots identifying ALDH1+ (CCSC) populations in xenograft tumors with Lnc34a knockdown (**H**) or ectopic Lnc34a expression (**I**). Scale bar, 50 μm. Error bars denote s.d. of triplicates. **p<0.01; ***p<0.001. p-value was calculated based on Student's t-test.

expression abrogates the effect of ectopic Lnc34a on tumor growth, resulting in similar tumor sizes as the control group (*Figure 3F,G*). Furthermore, we performed FACS on disassociated xenograft tumor cells. Lnc34a knockdown decreased the ALDH1+ CCSC population in the xenograft tumors (*Figure 3H*), while ectopic Lnc34a enriched the ALDH1+ CCSC population in the tumors (*Figure 3I*). Taken together, Lnc34a contributes to CCSC self-renewal and tumorigenesis.

## Lnc34a suppresses miR-34a expression

Opposite to Lnc34a, miR-34a is downregulated in ALDH1+ CCSCs and upregulated in ALDH1- non-CCSCs (*Figure 4A,B*). Knockdown of Lnc34a significantly increased miR-34a expression levels, while ectopic Lnc34a expression decreased miR-34a levels (*Figure 4C,D*). Therefore, Lnc34a suppresses miR-34a expression. RNA FISH showed that Lnc34a and miR-34a are mutually exclusive in the same daughter compartment and are present in opposite daughter compartments in more than 70% of CCSC1 and around 80% of CCSC2 dividing pairs (*Figure 4E,F*). On the other hand, we only observed symmetric distribution of p53, the other miR-34a upstream regulator (*Figure 4—figure supplement 1*). Therefore, Lnc34a provides a potential mechanism that accounts for asymmetric miR-34a levels in daughter pairs.

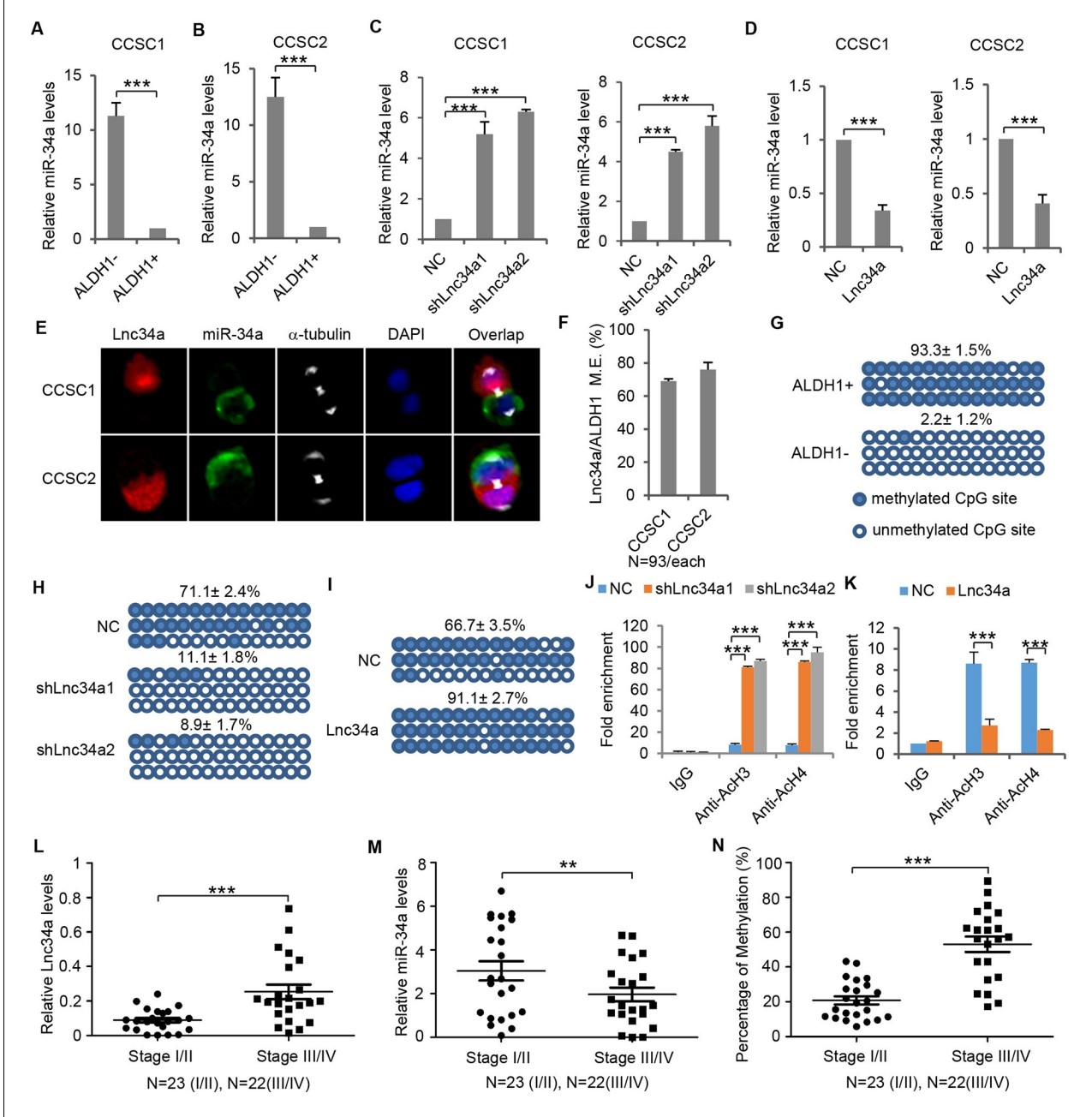

**Figure 4.** Lnc34a epigenetically silences miR-34a promoter. (A, B) RT-qPCR of miR-34a levels in CCSC1 (A) and CCSC2 (B). (C, D) RT-qPCR of miR-34a levels in CCSC1 (C) and CCSC2 (D) spheres with Lnc34a knockdown (shLnc34a1 and shLnc34a2) or ectopic expression (Lnc34a). NC is the control vector. (E) Representative images of Lnc34a and miR-34a asymmetry in CCSC1 and CCSC2 sphere cells. (F) Quantification of (E). Lnc34a and miR-34a distributions are mutually exclusive (M.E.) during most CCSC divisions. (G) Bisulfite sequencing analysis showing miR-34a promoter methylation status in ALDH1+ (CCSC) and ALDH1- (non-CCSC) cells isolated from sphere cells. PCR products amplified from bisulfite-treated genomic DNA were cloned and sequenced to reveal the methylation status of individual CpG sites. Percentages of the methylated CpG sites (filled circles) among all scored sites are indicated. (H) Lnc34a knockdown decreased miR-34a promoter methylation in sphere cells. (I) Ectopic Lnc34a expression increased miR-34a promoter methylation in sphere cells. (J, K) ChIP-qPCR with antibodies against acetylated histones H3 and H4. Lnc34a knockdown decreased miR-34a promoter acetylation (J), while ectopic Lnc34a expression increased acetylation (K). (L) RT-qPCR measurements of Lnc34a expression in early- and late-stage CRC specimens. (M) RT-qPCR measurements of miR-34a expression in early- and late-stage CRC specimens. (N) Bisulfite sequencing analysis of miR-34a promoter methylation status in early- and late-stage CRC specimens. Scale bar, 8 μm. Error bars denote s.d. of triplicates. **p<0.01; ***p<0.001. p-value was calculated based on Student's t-test.

The following figure supplements are available for figure 4:

*Figure 4 continued on next page*

*Figure 4 continued*

**Figure supplement 1.** p53 symmetry.
**Figure supplement 2.** Lnc34a epigenetically silences miR-34a promoters in Caco-2 and HT29 cells.
**Figure supplement 3.** Lnc34a, miR-34a, and promoter methylation levels in CRC specimens.

Bisulfite sequencing was then performed to evaluate miR-34a promoter methylation in ALDH1+ CCSCs and ALDH1- non-CCSCs isolated from spheres. 93.3% of tested CpG islands were methylated in CCSCs; in contrast, methylation rate was as low as 2.2% in non-CCSCs (*Figure 4G*). Knockdown of Lnc34a diminished overall miR-34a promoter methylation in sphere cells (*Figure 4H*), whereas ectopic Lnc34a expression significantly enhanced miR-34a promoter methylation, compared with the control vector (*Figure 4I*). Besides methylation, ChIP-qPCR showed that Lnc34a knockdown increases acetylated histones H3 and H4 associated with the miR-34a promoter (*Figure 4J*), whereas ectopic Lnc34a expression decreased acetylated histones H3 and H4 (*Figure 4K*). Taken together, the data suggests that Lnc34a silences miR-34a expression in CCSCs by promoting methylation and histone deacetylation of the miR-34a promoter. The effect of ectopic Lnc34a suggests that Lnc34a might act both in *cis* and in *trans*, as have been observed for various lncRNAs such as Evf-2 and the capacity of ectopically supplied *cis*-acting lncRNAs to act in *trans* (*Di Ruscio et al., 2013*; *Feng et al., 2006*; *Gomez et al., 2013*; *Jeon and Lee, 2011*; *Martianov et al., 2007*; *Rinn and Chang, 2012*; *Schmitz et al., 2010*).

Lnc34a also silences miR-34a in common CRC cell lines. Ectopic Lnc34a expression suppressed miR-34a expression, and promoted methylation and deacetylation of the miR-34a promoter in CRC cell lines Caco-2 and HT29 (*Figure 4—figure supplement 2*).

## Lnc34a, miR-34a, and promoter methylation are correlated with CRC progression

RT-qPCR performed in 23 early-stage (stage I/II) and 22 late-stage (stage III/IV) CRC specimens showed that Lnc34a expression is correlated with CRC progression. Overall, Lnc34a expression is lower in early-stage CRC and increases in late-stage CRC (*Figure 4L*, *Figure 4—figure supplement 3A*). miR-34a expression follows a reverse trend (*Figure 4M*, *Figure 4—figure supplement 3A*). Consistent with Lnc34a methylation of the miR-34a promoter, bisulfite sequencing revealed that the miR-34a promoter is more methylated in late-stage CRC than in early-stage CRC (*Figure 4N*, *Figure 4—figure supplement 3B*).

## Lnc34a interacts with epigenetic regulators

To understand the mechanisms via which Lnc34a regulates miR-34a expression, we performed an RNA pull-down assay with biotin-labeled Lnc34a, followed by mass spectrometry (MS), to search for potential Lnc34a-associated proteins. The DNA methyltransferase Dnmt3a, Histone Deacetylase 1 (HDAC1), and Prohibitin 2 (PHB2) were identified to be associated with Lnc34a (*Figure 5A* and *Figure 5—source data 1*). RNA immunoprecipitation (RIP) using specific antibodies against Dnmt3a, HDAC1 and PHB2 further confirmed the interactions (*Figure 5B*). In contrast, RNA pulldown and RIP did not detect any interaction between Lnc34a and Dnmt1, an enzyme that plays important roles in maintaining methylation during DNA replication (data not shown).

To investigate how Lnc34a interacts with Dnmt3a, HDAC1 and PHB2, we performed RIP while knocking down each of the proteins. Knockdown of PHB2 abolished the interaction between Lnc34a and Dnmt3a, but had no effect on the interaction between Lnc34a and HDAC1 (*Figure 5C*). Knockdown of Dnmt3a did not affect the interaction of Lnc34a with either PHB2 or HDAC1 (*Figure 5D*). Knockdown of HDAC1 did not interrupt Lnc34a and Dnmt3a interaction, and only had limited effect on Lnc34a and PHB2 interaction (*Figure 5E*). These data suggest that Lnc34a interacts with PHB2 and HDAC1, and recruits Dnmt3a through PHB2.

We then serially truncated Lnc34a and performed RNA pull-down assays to map HDAC1 and PHB2 binding to Lnc34a. The 1–267 bp fragment is sufficient to bind HDAC1, and the 560–693 bp fragment is sufficient to bind PHB2 (*Figure 5F*). Interaction between the fragments and their

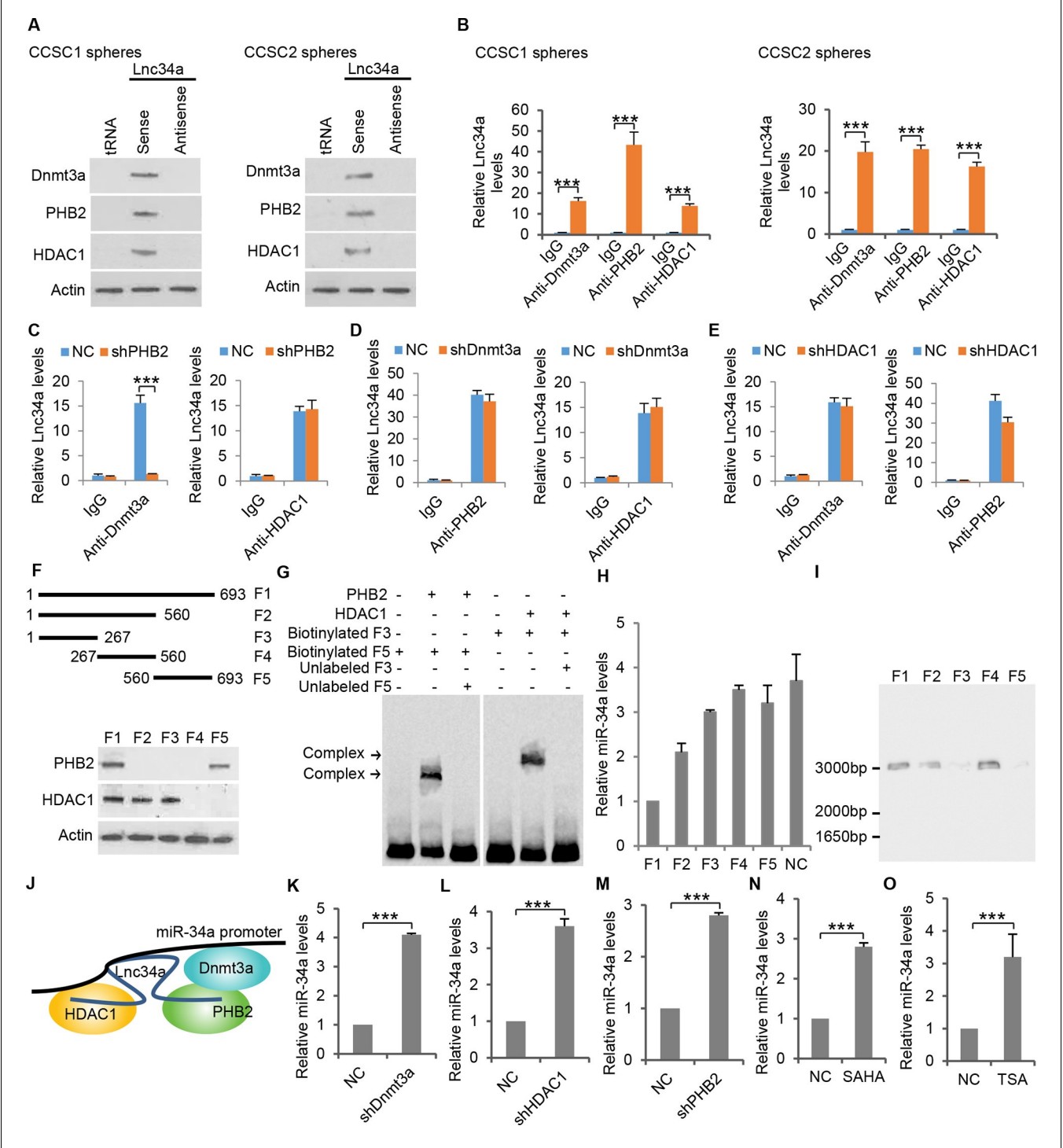

**Figure 5.** Lnc34a recruits epigenetic regulators. (**A**) Western blot following RNA-pull down showing Lnc34a interaction with PHB2, Dnmt3a and HDAC1 in CCSC1 (left) and CCSC2 (right) sphere cells. RNA-pull down was performed using CCSC lysates with biotin-labeled Lnc34a, antisense and tRNA. Actin was used for input control. (**B**) RNA immunoprecipitation (RIP) showing Lnc34a interaction with PHB2, Dnmt3a and HDAC1 in CCSC1 (left) and CCSC2 (right) sphere cells. (**C**) RIP showing PHB2 knockdown disrupts Lnc34a interaction with Dnmt3a, but has no effect on Lnc34a interaction with HDAC1. (**D**) RIP showing Dnmt3a knockdown does not affect Lnc34a interaction with PHB2 or HDAC1. (**E**) RIP showing HDAC1 knockdown has limited effect on Lnc34a interaction with PHB2 or Dnmt3a. (**F**) Mapping PHB2 and HDAC1 interaction domains on Lnc34a. Upper panel, schematic illustration of full-length Lnc34a and the truncated fragments for RNA put-down. Lower panel, Western blot of PHB2 and HDAC1 from RNA put-down of the fragments. (**G**) EMSA showing Lnc34a/PHB2 (left) and Lnc34a/HDAC1 (right) interactions. (**H**) RT-qPCR of miR-34a levels after expressing full-length or truncated fragments of Lnc34a. (**I**) In vitro interaction assay binding of the truncated fragment (267–560 bp) to the DNA containing the miR-34a

*Figure 5 continued on next page*

Figure 5 continued

promoter sequence. (J) Schematic illustration of Lnc34a interaction with PHB2, Dnmt3a and HDAC1. (K, L, M) RT-qPCR showing knockdown of Dnmt3a (K), HDAC1 (L), and PHB2 (M) increased miR-34a expression in sphere cells. (N, O) RT-qPCR showing treatments with HDAC inhibitor SAHA (N) or TSA (O) increased miR-34a expression in sphere cells. Error bars denote s.d. of triplicates. ***p<0.001. p-value was calculated based on Student's t-test.

The following source data is available for figure 5:

**Source data 1.** Potential Lnc34a-associated proteins identified by biotinylated Lnc34a pull-down and mass spectrometry.

cognate proteins were further validated by the electrophoretic mobility shift assay (EMSA). Incubation of labeled RNA probes Lnc34a:1–267 bp with recombinant HDAC1 and Lnc34a:560–690 bp with recombinant PHB2 resulted in specific gel retardation, while unlabeled RNA probes of the same fragments competitively disrupted those binding (*Figure 5G*). All three fragments are needed for full suppression of miR-34a expression (*Figure 5H*). Although the 267–560 bp fragment does not interact with either HDAC1 or PHB2, the in vitro interaction assay shows that it directly binds to the miR-34a promoter (*Figure 5I*). Therefore, Lnc34a binds to the miR-34a promoter via the 267–560 bp sequence, and recruits HDAC1 and Dnmt3a/PHB2 via the two flanking (1–267 bp and 560–690) sequences (*Figure 5J*).

We then knocked down PHB2, Dnmt3a and HDAC1 respectively, followed by RT-qPCR measurements of miR-34a expression. Knockdown of PHB2, Dnmt3a or HDAC1 upregulated miR-34a expression (*Figure 5I–K*). Inhibition of HDAC activity by SAHA and TSA also increased miR-34a expression (*Figure 5L,M*). The data suggest that these epigenetic regulators influence miR-34a expression levels.

## Discussion

The abundance of lncRNA in the human genome is being increasingly appreciated, but our understanding of their diverse functions is still lagging (*Mercer et al., 2009*; *Rinn and Chang, 2012*). We demonstrate that a lncRNA, Lnc34a, can initiate CCSC asymmetric division by targeting miR-34a. Previously, lncRNAs like HOTAIR and Xist have been shown to cause histone H3 lysine 27 methylation or lysine 4 demethylation (*Gupta et al., 2010*; *Tsai et al., 2010*; *Zhao et al., 2008*). Here, Lnc34a binds to the miR-34a promoter via its middle fragment, and recruits PHB2/Dnmt3a and HDAC1 via its flanking sequences to methylate and deacetylate the promoter, silencing miR-34a expression. This process reminds us of the ordered steps of protein-mediated DNA methylation—a DNA binding protein first interacts with the promoter, via which DNA methyltransferases are further recruited (*Chu et al., 2015*; *Serra et al., 2014*; *Wajapeyee et al., 2013*).

Lnc34a promotes CCSC self-renewal, and Lnc34a asymmetry leads to cell fate asymmetry in CCSC division. This effect is mediated by miR-34a, which has been shown to target factors of Notch and Wnt signaling pathways, both of which are essential for CCSC self-renewal (*Bu et al., 2013*; *Chen et al., 2015*; *Vermeulen et al., 2010*). In late-stage CRC, Lnc34a expression and miR-34a promoter methylation is upregulated, while miR-34a expression is downregulated. Lnc34a demonstrates that lncRNA can target microRNA for cellular control. Given that lncRNAs occupy the majority of the genome (*Mattick and Rinn, 2015*), lncRNA/microRNA circuitry can potentially increase the complexity of regulatory networks.

p53 is a well-known upstream regulator of miR-34a, and loss of p53 function certainly downregulates miR-34a. However, the discovery of Lnc34a demonstrates an alternative, epigenetic mechanism that cancer cells can utilize to silence miR-34a without having to mutate p53. Although p53 knockout has been reported to reduce asymmetric division in mammary stem cells (*Cicalese et al., 2009*), p53 is not known to be a major regulator of differentiation and is symmetric during CCSC division. Lnc34a provides normal and cancer cells a way to decouple mir-34a mediated cell fate decisions from p53, which may be present in both undifferentiated and differentiated cells.

## Materials and methods

### CCSC culture and sphere formation analysis

Human CRC cell lines Colo205, SW480, HT29, SW620, LS174T, DLD1, Caco-2 were purchased from ATCC and cultured in RPMI-1640 medium. No mycoplasma contamination was detected. Human CCSCs were isolated and cultured as described previously (*Bu et al., 2013*). Briefly, CCSCs were isolated from patient tumors by FACS based on markers CD44, CD133 and ALDH1 and functionally validated by serial sphere formation, tumor initiation, and self-renewal assays. For this study, original frozen stocks for the first passage were used. The CCSCs have not been authenticated by STR profiling. No mycoplasma contamination was detected. CCSCs were cultured as spheres in ultralow-attachment flasks (Corning, Tewksgury, MA) in DMEM/F12 (Invitrogen, Pittsburg, PA), supplemented with nonessential amino acids (Fisher, Pittsburg, PA), sodium pyruvate (Fisher), Penicillin-streptomycin (Fisher), N2 supplement (Invitrogen), B27 supplement (Invitrogen), 4 µg/mL heparin (Sigma, Mendota Heights, MN), 40 ng/mL epidermal growth factor (Invitrogen), and 20 ng/mL basic fibroblast growth factor (Invitrogen) at 37°C and 5% $CO_2$.

To measure tumor sphere formation, single CCSCs were plated in 24-well ultra-low attachment plates (Corning) at 1,000 cells per well. Tumor spheres were counted after 2 weeks in culture by an inverted microscope (Olympus).

### Clinical specimens

45 frozen CRC specimens of different clinical stages were acquired from Weill Cornell Medical College (WCMC) Colon Cancer Biobank. The CRC stage was determined according to the TNM staging system. The clinical data for the patients are summarized in *Figure 1—source data 1*. The studies followed informed consent and approval of the IRB committee at Weill Cornell Medical College.

### Immunofluorescence

Pair-cell assay for CCSC division were performed as described previously (*Bultje et al., 2009*). Briefly, spheres were dissociated and the single cells were plated on an uncoated glass culture slide (Corning) and allowed to divide once. After being fixed and blocked, the cells were incubated with anti-ALDH1 (clone H-4, 1:100, Santa Cruz, Dallas, TX), anti-CD133 (1:200, Abcam, Cambridge, MA) and anti-α-tubulin (1:500, Abcam) antibodies overnight at 4°C. For the BrdU incorporation assay, sphere cells were cultured in proliferative medium (DMEM with 10% FBS) for 24 hr. Single cells were then plated and allowed to divide once in proliferative medium (1st division). After treatment with BrdU (Sigma) for 3 hr, the cells were fixed in cold 70% ethanol, incubated in 2 M HCl for 1 hr, washed, and switched to 100 mM $Na_2B_4O_7$ for 2 min. After being blocked in 10% normal goat serum for 1 hr, the cells were then incubated with anti-BrdU (1:200, Sigma) antibody at 4°C overnight. The cells were then incubated with fluorescence-conjugated secondary antibody or streptavidin (Invitrogen) for 1 hr at room temperature. After counterstained with DAPI (Invitrogen), the slides were observed under a fluorescent microscope (Olympus, Jupiter, FL).

### RNA FISH

RNA FISH was performed as described previously (*Lu and Tsourkas, 2009*). In this study, Digoxigenin (DIG)–labeled locked nucleic acid (LNA) probe (Exiqon, Woburn, MA) against miR-34a or Biotin-labeled LNA probe against Lnc34a (Exiqon) were used for RNA FISH. RNA expression was detected by Rhodamine Red labeled secondary antibody or Alexa Fluor 488 conjugated streptavidin (Invitrogen). Anti-α-tubulin was used to identify dividing cells and DAPI (Invitrogen) was used for nucleic counterstaining.

### Lnc34a cloning, shRNAs, northern blot and bisulfite sequencing

A 293 bp fragment was amplified using primers: 5'-GGTGGAGGAGATGCCGC-3' and 5'-ACCTGGG TGCATGCTGGGACG-3'. To identify the full length of Lnc34a, 3'RACE and 5'RACE was performed using kit with the primers: 5'- GCAGGACTCCCGCAAAATCTC-3' and 5'- CTCAGTCCGTGCGAAAG TTTG-5' respectively. The full length of Lnc34a was then amplified using the primers: 5'-TTAACCAG TCGGCCTTCCTCGCC-3' and 5'-TGAGATTAACCGACTTTCCCAAG-3', then cloned into pGEM-T (Promega, Durham, NC) for sequencing. The full length of Lnc34a was cloned into pMSCV PIG

vector (Addgene, Cambridge, MA) for ectopic Lnc34a expression study. shRNAs against Lnc34a were designed using Invitrogen online tool and cloned in pMSCV PIG vector. shRNAs against PHB2, Dnmt2a and HDAC1 were purchased from Sigma. The knockdown efficiency was validated by RT-qPCR. Northern blot was performed using NorthernMax Kit (Invitrogen) according to the manufacturer's instructions. The probes were generated using PCR DIG Probe Synthesis Kit (Roche, Indianapolis, IN) with the primers: 5'- TAGCCGAGCAAAACCCC-3' and 5'- ATGTGGGA-CACGGATGAGA-3'. Bisulfite sequencing was performed using EZ DNA methylation kit (Zymo, Irvine, CA). 9 sequencing runs were carried out for each condition.

### Flow cytometry

Flow cytometry were performed as described previously (4). CD133 expression was detected using anti-CD133 (clone C24B9, 1:50, Cell Signaling, Beverly, MA) and ALDH1 levels were analyzed using the Aldeflour kit. The samples were analyzed using a BD LSR II flow cytometer. The raw FACS data were analyzed with the FlowJo software to gate cells according to their forward (FSC) and side (SSC) scatter profiles.

### Quantitative real-time RT-PCR analysis

Total RNA was extracted from the cells using the TRIzol Reagent (Invitrogen). cDNA was synthesized using the High Capacity cDNA Archive Kit (Applied Biosystems, Foster city, CA). Quantitative PCR was carried out using the TaqMan MicroRNA Assay (Applied Biosystems) to detect miR-34a levels and the SYBR Green System (Applied Biosystems) to detect other gene expression. The miR-34a primer and U6 primer were purchased from Applied Biosystems. Other primer sequences include: Lnc34a, 5'-GGAGGCTACACAATTGAACAGG-3' and 5'-AGTCCGTGCGAAAGTTTGC-3'; actin, 5'-CGCGAGAAGATGACCCAGAT-3' and 5'-ACAGCCTGGATAGCAACGTACAT-3';. The expression of each gene was defined from the threshold cycle (Ct), and the relative expression levels were calculated using the 2-$\triangle\triangle$Ct method after normalization to the actin expression level.

### RNA pull-down assay, mass spectrometry, and electrophoretic mobility shift assay (EMSA)

Full length of Lnc34a cDNA and it truncations were cloned into pGEM-3ZF(+). Biotin-labeled RNAs were transcribed from the linearized pGEM-3ZF plasmid in vitrousing a biotin labeling mix (Roche) and T7 polymerase (Promega). The biotinylated RNA was heated to 90°C for 2 min, incubated on ice for 2 min, and then shifted to RT for 20 min with RNA renature buffer (10 mM tris-HCL pH7.0, 0.1M KCL, 10 mM MgCl2 to allow proper secondary structure formation. The cell lysates were freshly prepared using RIPA buffer (Millipore, Billerica, MA) with proteinase inhibitor (Roche). After precleared using Dynabeads M-270 streptavidin (Invitrogen), the cell lysates were diluted in binding buffer and incubated with the folded RNA for 2 hr at 4°C. Dynabeads M-270 streptavidin were then added into the mixture and incubated for 1 hr at 4°C. After washing, the RNA-binding protein complexes were released from the Dynabeads. The retrieved proteins were collected for Mass Spec and Western blotting validation. RNA-EMSA was performed using a LightShift Chemiluminescent RNA EMSA Kit (Thermo Scientific, Pittsbrugh, PA) according to the manufacturer's instructions.

### RNA immunoprecipitation (RIP) and chromatin immunoprecipitation (ChIP) assays

RIP assays were performed using a RIP RNA-binding protein immunoprecipitation kit (Millipore) according to the manufacturer's instructions. Antibodies against PHB2 (Bethl, Montgomery, TX), HDAC1 (Bethl) and Dnmt3a (Abcam) were added into the cell lysates. Lnc34a was retrieved from the complexes and evaluated by RT-qPCR. ChIP was performed using a ChIP assay kit (Millipore) as described previously (4). Antibodies against acetylated histones H3 and H4 (Millipore) were used to evaluate histone modifications associated with the miR-34a promoter. Enrichment of miR-34a promoter fragments was quantified by RT-qPCR with the primers: 5'-CACCTGGTCCTCTTTCCTTT-3' and 5'- TCCTCCTTCCTGCTCGT -3'.

## Western blot

Cells were lysed in RIPA lysis buffer supplemented with cocktail protease inhibitor (Roche). Proteins were separated by SDS-PAGE and transferred onto a Hybond membrane (Amersham). The membranes were incubated with primary antibodies either anti-PHB2 (1:1000, Bethl), anti-Dnmt3a (1:500, Abcam), anti-HDAC1(1:1000, Bethl) or anti-Actin (1:1000, Abcam) in 5% milk/TBST buffer (25 mM Tris pH 7.4, 150 mM NaCl, 2.5 mM KCl, 0.1% Triton-X100) overnight, followed by incubation with horseradish peroxidase (HRP)-conjugated anti-mouse or anti-rabbit IgG (Santa Cruz) for 1 hr. The target proteins were detected on membrane by enhanced chemiluminescence (Pierce, ).

## Statistical analysis

Data were expressed as mean ± standard deviation of three biological repeats. Student t-tests were used for comparisons, with $p<0.05$ considered significant.

## Acknowledgements

This work was supported by NIGMS R01GM95990, R01GM114254, NSF 1350659, R01 Ca098626, NSF 1137269, DARPA 19–1091726, and NYSTEM C029543.

# Additional information

## Funding

| Funder | Grant reference number | Author |
| --- | --- | --- |
| National Science Foundation | 1350659 | Xiling Shen |
| National Institutes of Health | R01GM114254 | Xiling Shen |
| National Institutes of Health | R01GM95990 | Xiling Shen |

The funders had no role in study design, data collection and interpretation, or the decision to submit the work for publication.

## Author contributions

LW, PB, Performed the molecular experiments and some of the animal-related experiments, Conception and design, Analysis and interpretation of data, Drafting or revising the article; YA, Helped LW and PB with the molecular experiments and some of the animal-related experiments; TS, Performed immunofluorescence on the tissue samples; HJC, KX, Performed the rest of the animal-related experiments; SML, Contributed clinical samples, Conception and design, Analysis and interpretation of data, Drafting or revising the article; XS, Conception and design, Analysis and interpretation of data, Drafting or revising the article

## Author ORCIDs

Xiling Shen, http://orcid.org/0000-0002-4978-3531

## Ethics

Human subjects: Frozen CRC specimens of different clinical stages were acquired from Weill Cornell Medical College (WCMC) Colon Cancer Biobank. The studies followed informed consent and approval of the IRB committee at Weill Cornell Medical College.

Animal experimentation: All animal experiments were approved by The Cornell Center for Animal Resources and Education (CARE) and followed the protocol (2009-0071 and 2010-0100).

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
