## [Decision Letter]

Thank you for submitting your work entitled "A Long Non-Coding RNA Targets MicroRNA miR-34a to Regulate Colon Cancer Stem Cell Asymmetric Division" for consideration by *eLife*. Your article has been favorably evaluated by Kevin Struhl (Senior editor) and three reviewers, one of whom, Michael Green, is a member of our Board of Reviewing Editors, and another is Narendra Wajapeyee.

The reviewers have discussed the reviews with one another and the Reviewing Editor has drafted this decision to help you prepare a revised submission.

Summary:

In this study, Wang et al. report that a novel lncRNA, lnc34a, mediates epigenetic silencing of the microRNA, mir-34a, in colorectal cancer stem cells (CCSCs). Epigenetic silencing is due to the ability of lnc34a to recruit PHB2, HDAC1 and DNMT3a. The authors show that Lnc34a levels affect CCSC self-renewal and colorectal cancer (CRC) growth in xenograft models. In addition, Lnc34a is upregulated in late-stage CRCs, contributing to epigenetic miR-34a silencing and CRC proliferation. Finally, the authors report that Lnc34a is initiates asymmetric cell division of CCSCs and is asymmetrically distributed.

General Assessment:

In general, this is a very interesting study whose conclusions are supported by the presented results. However, as described below, there are several additional experiments and analyses that could be performed to strengthen the main conclusions.

Essential revisions:

The authors are encouraged to directly address as many of the following points as possible.

1) Figure 1. The increased levels of lnc34a in CCSCs appear to be greater when by qRT-PCR than by northern blotting. If this is correct, the authors should comment on this difference. In any case, the authors should provide quantification of the northern blots.

In Figure 1—figure supplement 1, can the authors provide northern blots for lnc34a in the other 8 CRC cell lines?

2) RNA FISH experiments. For all RNA FISH experiments, some type of quantification similar to that in Figure 1 needs to be provided. Showing a representative example is not sufficient.

3) In bisulfite sequencing experiments it is typical to show 6-10 independent sequences, whereas in this study only 3 sequences are provided.

4) The authors can substantially strengthen their main conclusions by performing a rescue experiment to confirm that mir34a is the critical target of lnc34a. For example, ectopically express lnc34a in an appropriate cell type and show that the phenotype can be reversed by ectopically expressing mir34a. For example, this could be done using the cell type and readout in Figure 3. How do the authors envision that lnc34a is targeted to mir34a? Do they think this is by base-pairing or RNA loop formation? Any experimental results on this question would provide deeper understanding of how lncRNAs mediate epigenetic silencing. An initial experiment might be to ectopically express lnc34a truncation derivatives (see Figure 5) to identify the region(s) of lnc34a required for silencing.

Optional actions and minor points:

1) Do the authors think that mir34a is the sole target that is epigenetically silenced by lnc34a or might other genes be epigenetically silenced by lnc34a. The authors could address this question by ectopically expressing lnc34a in an appropriate cell type followed by RNA-Seq analysis.

2) What do the authors think is the mechanistic basis for the asymmetric distribution of lnc34a following cell division (see Figure 2).

3) Can the authors comment on which mRNA miR-34a is antagonizing in the CCSC? For example, are there mRNAs that decreased when mir-34a is present?

4) It was surprising that the authors did not discuss or present evidence for WNT signaling activity and its role in potentially mediating the effects of Lnc34a. A very important previous study has shown that WNT activity defines colon cancer cell stems and shows that in part it was a paracrine effect (Vermeulen et al., Nature Cell Biology, 12, 468-476, 2010). It will be important to put the current study in the context of previous observations and it will be important to measure the WNT signaling markers after altering the expression of Lnc34a and miR-34a.

5) Does miR-34a induction by p53 modulate the expression level of Lnc34a (e.g., repression of Lnc34a by promoter interference)?

6) Ectopic expression of Lnc34a is used in several key experiments. The authors need to confirm nuclear localization of ectopic Lnc34a.

7) Can the authors please comment on whether or not there are previous examples of lncRNAs that mediate epigenetic silencing.

8) The authors provided solid evidence for a model of ordered recruitment of epigenetic regulators in which the final step is recruitment of a DNA methyltransferase. Work from Michael Green's laboratory has also shown that epigenetic silencing occurs by an ordered pathway of recruitment on the promoter and the final step is recruitment of a DNA methyltransferase (see for example Wajapeyee, G&D, 2013; Serra et al., *eLife*, 2014; Fang et al., Mol Cell, 2014). It would be useful to mention the similar findings in very different experimental systems in the Discussion.

9) It is also notable that the authors find that DNMT3a is the DNA methyltransferase involved in mir34a silencing. It has been generally believed that only DNMT1 can carry out maintenance methylation, which is required for the maintenance of epigenetic silencing. However, here it appears that DNMT3a is functioning as a maintenance DNA methyltransferase. The authors may want to strengthen this conclusion by showing that DNMT1 knockdown doesn't reactivate mir34a and/or the DNMT1 is not recruited by lnc34a. If true, the authors should point this out in the Discussion section. Again, similar findings on the role of maintenance DNA methyltransferases other than DNMT1 have been reported by the Green lab (see, for example, Fang et al., Mol Cell, 2014).

10) In the subsection “Quantitative real-time RT-PCR analysis”: Ln34a should be Lnc34a.

11) In the subsection “Western blot”: anti-Action should be anti-Actin.

12) Can Lnc34a be detected in publicly available RNA-Seq dataset in human cell lines?

---

## [Author Response]

*Essential revisions: The authors are encouraged to directly address as many of the following points as possible. 1) Figure 1. The increased levels of lnc34a in CCSCs appear to be greater when by qRT-PCR than by northern blotting. If this is correct, the authors should comment on this difference. In any case, the authors should provide quantification of the northern blots.* The reviewers are correct. We quantified Northern blotting using Image J (Figure 1). The relative amounts of lnc34a in HT29, Caco-2, CCSC1 and CCSC2 are 1, 2.55, 9.4, 6.98. On the other hand, RT-qPCR indicated the relative amounts of lnc34a as 1, 2.5, 18.2 and 16.1. We think the discrepancy is likely due to signal saturation of lnc34a in CCSCs while we were trying to capture low lnc34a signals in HT29 cells during Northern blot. Notably, in HT29 and Caco-2 cells, which express low level of lnc34a, the lnc34a levels measured by Northern blot were consistent with those measured by RT-qPCR.

*In Figure 1—figure supplement 1, can the authors provide northern blots for lnc34a in the other 8 CRC cell lines?* We performed Northern blot for lnc34a in the 9 cell lines showed in Figure 1—figure supplement 1.

*2) RNA FISH experiments. For all RNA FISH experiments, some type of quantification similar to that in Figure 1 needs to be provided. Showing a representative example is not sufficient.* We provided the quantifications for the immunofluorescent imaging experiments (Figure 2—figure supplement 2C, Figure 4—figure supplement 4B and 4D).

*3) In bisulfite sequencing experiments it is typical to show 6-10 independent sequences, whereas in this study only 3 sequences are provided.* We apologize for the confusion. For each condition, we performed bisulfite conversion 3 times independently. Each time, we picked 3 clones for bisulfite sequencing. Therefore, we performed 9 independent sequencing experiments for each condition, not 3. We clarified it in the Methods and included s.d. in the figures (Figure 4).

4) The authors can substantially strengthen their main conclusions by performing a rescue experiment to confirm that mir34a is the critical target of lnc34a. For example, ectopically express lnc34a in an appropriate cell type and show that the phenotype can be reversed by ectopically expressing mir34a. For example, this could be done using the cell type and readout in Figure 3. How do the authors envision that lnc34a is targeted to mir34a? Do they think this is by base-pairing or RNA loop formation? Any experimental results on this question would provide deeper understanding of how lncRNAs mediate epigenetic silencing. An initial experiment might be to ectopically express lnc34a truncation derivatives (see Figure 5) to identify the region(s) of lnc34a required for silencing.

First, we performed the rescue and measured sphere formation, CCSC division and xenograft tumor formation. As shown in Figure 2 and Figure 3, ectopic miR-34a expression reversed the phenotype of ectopic lnc34a expression, confirming that miR-34a is the critical target of lnc34a.

Next, we measured miR-34a expression with ectopic expression of the lnc34a truncates, which showed that all three lnc34a truncates are required to suppress miR-34a expression (Figure 5). Then, we amplified the 3kb miR-34a promoter region and used biotin-labeled lnc34a truncates as probes to measure the interaction between lnc34a truncates and miR-34a promoter. As shown in Figure 5, only the fragment (267bp-560bp) can interact with miR-34a promoter. This interaction allows the recruitment of HDAC1 and Dnmt3a (via PHB2), which bind to the fragments (1bp-267bp) and (560bp-693bp) respectively (Figure 5). We thank the reviewers for these great suggestions, which help strengthen the paper significantly.

*Optional actions and minor points: 1) Do the authors think that mir34a is the sole target that is epigenetically silenced by lnc34a or might other genes be epigenetically silenced by lnc34a. The authors could address this question by ectopically expressing lnc34a in an appropriate cell type followed by RNA-Seq analysis.* miR-34a targets many important genes (e.g., BCL2, CD44, NOTCH1/2, DLL1, JAG1, Sirtuin 1, TCF, MYC, YY1, HDAC1, cyclin D1, CDK4/6, VEGFA, MAP2K1, HNF4a, etc.) to regulate diverse cellular processes including cell fate, cell cycle, apoptosis, senescence, metabolism, epigenetic regulation, and so on. Ectopic lnc34a suppresses miR-34a, hence altering all the miR-34a downstream pathways and suppressing CCSC differentiation. Therefore, it will be challenging to distinguish direct lnc34a targets from miR-34a downstream targets solely based on RNA-Seq data.

Although we cannot completely rule out the possibility of other lnc34a targets, it is highly likely that the miR-34a promoter is the only or at least the most significant target of lnc34a, because: 1. lnc34a is adjacent to and overlapping with the miR-34a promoter, and 2. our analysis in response to major concern 4 (Figure 5) showed that the middle lnc34a fragment binds to the miR-34a promoter with overlapping sequence while the two flanking fragments recruit epigenetic regulator.

*2) What do the authors think is the mechanistic basis for the asymmetric distribution of lnc34a following cell division (see Figure 2).* We hypothesize that the lnc34a promoter remains active in the CCSC daughter cell compartment but is immediately silenced in the non-CCSC daughter compartment. This can lead to asymmetric lnc34a levels given that lnc34a does not contain a poly-A tail and hence likely has short half-life. However, there could be alternative mechanisms such as lnc34a was actively recruited asymmetrically. We do not know the answer, and this will be an area for future investigation.

*3) Can the authors comment on which mRNA miR-34a is antagonizing in the CCSC? For example, are there mRNAs that decreased when mir-34a is present?* Notch and Wnt signaling have been shown to be essential for normal and colon cancer stem cell self-renewal. We have previously shown that miR-34a targets Notch signaling to promote CCSC differentiation (Bu et al., 2013). Interestingly, a recent paper showed that the Wnt signaling factor TCF is a direct target of miR-34a (Chen et al., 2015). To investigate whether miR-34a antagonizes TCF in CCSCs, we measured the TCF level by Western blot with ectopic expression of miR-34a and lnc34a. Ectopic lnc34a expression (miR-34a suppression) upregulated TCF expression, while ectopic miR-34a expression suppressed TCF expression (Figure 6). Therefore, miR-34a likely antagonizes both Notch and Wnt signaling to promote differentiation in CCSCs. We included the discussion in the revised manuscript.

Author response image 1.Lnc34a upregulates TCF1 via suppression of miR-34a.Western blot showing TCF1 was suppressed by ectopic miR-34a expression and upregulated by ectopic Lnc34a expression in CCSCs. Ectopic miR-34a abrogated the upregulation of Lnc34a on TCF1.**DOI:**
http://dx.doi.org/10.7554/eLife.14620.016

*4) It was surprising that the authors did not discuss or present evidence for WNT signaling activity and its role in potentially mediating the effects of Lnc34a. A very important previous study has shown that WNT activity defines colon cancer cell stems and shows that in part it was a paracrine effect (Vermeulen et al., Nature Cell Biology, 12, 468-476, 2010). It will be important to put the current study in the context of previous observations and it will be important to measure the WNT signaling markers after altering the expression of Lnc34a and miR-34a.* As described above in 3, we measured TCF by western blot when ectopically expressing lnc34a and miR-34a, which indicated that miR-34a inhibits TCF, while lnc34a upregulates TCF (Figure 6), consistent with recent reports showing TCF being a direct target of miR-34a (Chen et al., 2015). We included the discussion and the Vermeulen et al. and Chen et al. references in the revised manuscript.

*5) Does miR-34a induction by p53 modulate the expression level of Lnc34a (e.g., repression of Lnc34a by promoter interference)?*We ectopically expressed p53 in HCT116 cells (which has low lnc34a expression) and CCSCs (which has high lnc34a expression). RT-qPCR showed that ectopic p53 expression upregulates miR-34a expression in both cell lines. In contrast, ectopic p53 expression does not change Lnc34a expression significantly (Figure 7).

Author response image 2.Lnc34a and induction of miR-34a by p53 does not affect each other.(**A** and **B**) RT-qPCR showing miR-34a and Lnc34a expression level in HCT116 (**A**) and CCSC1 (**B**) cells with p53 ectopic expression.**DOI:**
http://dx.doi.org/10.7554/eLife.14620.017

*6) Ectopic expression of Lnc34a is used in several key experiments. The authors need to confirm nuclear localization of ectopic Lnc34a.* We performed RT-qPCR to measure cytoplasmic and nuclear lnc34a levels with ectopic expression of lnc34a. As shown in Figure 1—figure supplement 2, higher lnc34a level is detected in nucleus than in cytoplasm.

*7) Can the authors please comment on whether or not there are previous examples of lncRNAs that mediate epigenetic silencing.* There are several reports of lncRNAs mediating epigenetic silencing, including HOTAIR and Xist. HOTAIR interacting with PRC2 and LDS1mediates histone H3 lysine 27 methylation and lysine 4 demethylation (Gupta et al., 2010; Tsai et al., 2010). Xist binds to PRC2 to induce H3 lysine 27 methylation (Zhao et al., 2008). We have included these references in the revised manuscript.

*8) The authors provided solid evidence for a model of ordered recruitment of epigenetic regulators in which the final step is recruitment of a DNA methyltransferase. Work from Michael Green's laboratory has also shown that epigenetic silencing occurs by an ordered pathway of recruitment on the promoter and the final step is recruitment of a DNA methyltransferase (see for example Wajapeyee, G&D, 2013; Serra* et al.

*, eLife, 2014; Fang et al., Mol Cell, 2014). It would be useful to mention the similar findings in very different experimental systems in the Discussion.*

We have cited the mentioned papers and discussed this point in the revised manuscript.

*9) It is also notable that the authors find that DNMT3a is the DNA methyltransferase involved in mir34a silencing. It has been generally believed that only DNMT1 can carry out maintenance methylation, which is required for the maintenance of epigenetic silencing. However, here it appears that DNMT3a is functioning as a maintenance DNA methyltransferase. The authors may want to strengthen this conclusion by showing that DNMT1 knockdown doesn't reactivate mir34a and/or the DNMT1 is not recruited by lnc34a. If true, the authors should point this out in the Discussion section. Again, similar findings on the role of maintenance DNA methyltransferases other than DNMT1 have been reported by the Green lab (see, for example, Fang et al., Mol Cell, 2014).* Our RNA pull down assay did not show DNMT1 interaction with lnc34a. To further confirm our observation, we performed RIP using DNMT1 antibody, and did not detect lnc34a either. Therefore, although Dnmt1 is important for maintaining DNA methylation during DNA replication, lnc34a may recruit DNMT3a to the miR-34a promoter for de novo methylation. We cited the above paper and discussed this point in the revised manuscript.

10) In the subsection “Quantitative real-time RT-PCR analysis”: Ln34a should be Lnc34a.

We have corrected this issue in the revised manuscript.

11) In the subsection “Western blot”: anti-Action should be anti-Actin.

We have corrected this in the revised manuscript.

12) Can Lnc34a be detected in publicly available RNA-Seq dataset in human cell lines?

We have checked public databases including TCGA. Lnc34a was not detected in the datasets, probably because lnc34a does not contain a poly-A tail while most of the RNA-Seq experiments focused on mRNA enriched by poly-A selection.